# The Effects of Providing Outdoor Access to Broilers in the Tropics on Their Behaviour and Stress Responses

**DOI:** 10.3390/ani12151917

**Published:** 2022-07-27

**Authors:** Rubí E. Sánchez-Casanova, Luis Sarmiento-Franco, Clive J. C. Phillips

**Affiliations:** 1Facultad de Medicina Veterinaria y Zootecnia, Universidad Autónoma de Yucatán, Km 15.5 Carretera Mérida-Xmatkuil, Apdo. 4-116, Itzimná, Mérida, Yucatan 97100, Mexico; luis.sarmiento@correo.uady.mx; 2Institute of Veterinary Medicine and Animal Sciences, Estonian University of Life Sciences, Kreutzwaldi 1, 51014 Tartu, Estonia; clive.phillips@curtin.edu.au; 3Curtin University Sustainability Policy Institute, Curtin University, Kent Street, Bentley, WA 6102, Australia

**Keywords:** outdoor access, stocking density, broiler welfare, behavior, stress, season, tropics

## Abstract

**Simple Summary:**

The effects of outdoor access for broilers have been tested under temperate conditions, where free-range systems have begun to be widely used. However, under tropical conditions, where the birds may be heat-stressed outside, the benefits of providing a free-range area may be less evident. We compared whether access to an outdoor area improved behavior and several physiological welfare indicators of broilers at two stocking densities in a tropical environment. There were no major effects of outdoor access on broiler behavior, except that resting was reduced by providing outdoor access to older birds and those at low stocking densities inside. However, outdoor access increased heterophil numbers in summer, but not in winter, which may indicate heat stress. We concluded that the effects of outdoor access on the welfare of broilers in the tropics are dependent on season and stocking density.

**Abstract:**

The effects of outdoor access, stocking density, and age on broiler behavior, stress, and health indicators in a tropical climate were assessed over two seasons, winter and summer. Two hundred and forty Cobb500 male chickens were allocated to one of four treatments, with six replicates of ten birds in each: low stocking density indoors with outdoor access (LO); high stocking density indoors with outdoor access (HO); low stocking density indoors without outdoor access (LI); and high stocking density indoors without outdoor access (HI). Scan sampling was used to record their behavior both indoors and outdoors. At 28 and 42 days old, blood samples were obtained to determine the heterophil to lymphocyte (H/L) ratio. At 42 days old, chickens were culled and inspected for footpad dermatitis (FPD), and bone quality was examined. Their spleens and bursas of Fabricius were collected and weighed, relative to carcass weight (RW). A factorial analysis was used to test the effects of season (winter or summer), outdoor access (with or without), stocking density (low: 5 animals/m^2^ or high: 10 animals/m^2^), and age (28 or 42 days) on the behavior and stress and health indicators. There were no major effects of providing outdoor access on behavior, except that resting was reduced by providing outdoor access to older birds and those at low stocking densities inside. Resting was also greater in indoor and high-density treatments during winter. The bursa of Fabricius was heavier in summer in outdoor birds. The tibia bones were shorter in the outdoor birds. Heterophil numbers were greater in the outdoor treatments in summer but not in winter. These results indicate that outdoor access can increase activity in some situations, and potentially increase bone strength, but it may also increase the risk of stress, particularly heat stress in summer.

## 1. Introduction

Broilers make a major contribution to human protein consumption in many regions of the world; to achieve this, the poultry industry has adopted high stocking densities to maximize output and profitability [1,2]. Compared with their wild ancestors, the ability of chickens to cope with the stressors in an outdoor environment appears to have decreased [3]. Temperatures above 30 °C and relative humidity levels above 80% inside the housing, as observed in tropical environments during summer, can cause heat stress in fast-growing chickens, which adversely affects growth rates, immune functions, disease susceptibility, and potentially death by heat exhaustion [4,5]. Coping strategies during stressful circumstances require major physiological and behavioral adjustments [6]. Physiological responses include an increase in the heterophil to lymphocyte (H/L) ratio, due to heterophilia and lymphopenia [7], and increased weight of lymphoid organs, through immune activation [8]. Both are good indicators of stress and immune status.

Behavioral responses, such as shade seeking or increased standing to allow increased ventilation, require the birds to walk, run, and stand. This could be dependent on leg health, which can be assessed by the Seedor index of bone density [9].

Despite these potential problems in the tropics, there is a growing body of literature that recognizes the importance of allowing chickens to express their natural behavior by providing outdoor access [10,11,12], particularly at low stocking densities [13]. Indoor stocking densities may influence whether outdoor access is beneficial or not. However, there have been few studies investigating the effects of offering outdoor access to broilers housed under tropical climatic conditions. In an earlier study of the provision of outdoor area for broilers in the winter in the tropics [13], we found that birds at a high stocking density with no outdoor pens walked and preened themselves less and laid down more. Outdoor access increased foraging but only when birds were stocked at a low stocking density indoors; these birds also appeared more responsive to stressors, with elevated corticosterone and reduced spleen and bursa of Fabricius weights. Therefore, the aim of this study was to evaluate, in two contrasting seasons of the year, the effects on behavior and stress responses to allow chickens outdoor access at two indoor stocking densities. The effects on bird growth and carcass composition were published previously [14], showing that the low stocking density and outdoor access had positive effects on chicken growth in winter. Summer temperatures were sufficient to cause heat stress, as evidenced by an increase in mortality. Birds reared in summer without outdoor access, or with outdoor access but a high stocking density inside, had reduced growth and feed conversion efficiency compared with birds reared in winter. Because of the evident effects of outdoor access on walking and lying that we previously recorded [13], the effects on leg health (footpad dermatitis and bone conformation) were included in this study.

## 2. Materials and Methods

The study was carried out at the research farm of the Universidad Autónoma de Yucatán (20°51′47″ N 89°37′19″ W) at an altitude of 10 m.a.m.s.l., with warm subhumid tropics, a total annual rainfall of 1100 mm, and a mean annual temperature of 28 °C, potentially increasing to 40 °C in spring and summer [13,15].

### 2.1. Experimental Treatments

In two contrasting seasons of the year, winter (19 December 2019–30 January 2020) and summer (22 June–3 August 2020), 240 male chickens (different birds in each season) were randomly assigned to one of four treatments in a two-factor (stocking density (low and high) and outdoor access (indoor only, hereafter indoor, and indoor with outdoor access, hereafter outdoor)) factorial design: low stocking density with outdoor access (LO); high stocking density with outdoor access (HO); low stocking density indoors without outdoor access (LI); and high stocking density indoors without outdoor access (HI). Each treatment had six replicates. The 60 birds in each treatment were allocated to six groups of ten birds [13]. Two different enclosure sizes were used to achieve the two stocking densities, with a constant group size, as the latter could be confounded with the otal area available for the group [14].

### 2.2. Housing and Management

One-day-old Cobb500 male chicks, obtained from a commercial hatchery, were placed inside a circular reception area made with cardboard walls. They were vaccinated against Gumboro and Newcastle at 7 and 21 days old, respectively [14]. The lighting program was 24L:0D for the first seven days. One week later (at seven days of age), ten birds were randomly allocated to each of 2the 4 naturally ventilated pens. After an adaptation period of 7 days (i.e., when the birds were 14 days old), outdoor access was provided for birds in the LO and HO treatments [13]. The artificial lighting program in the indoor facility was 16L:8D from day 8 to 14, 17L:7D from day 15 to 21, and 14L:10D from day 22 to the end of the experiment. On average, 11 h of natural light (06:00–17:00 h) was provided daily, and the remaining light hours were provided by using one 40 W incandescent bulb (15 lumens/watt), providing 17 lux in each of the 3 buildings used for the experiment. Pens were arranged inside three portal-framed buildings of 6 m × 6 m to minimize position effects in both horizontal directions, although the treatments with outdoor access were necessarily on one side of the building. Pens were constructed of galvanized chicken wire mesh to a height of 1.20 m with wood shavings as the bedding material. In each of the three buildings, there were eight pens [14].

Feed and water were provided *ad libitum* through hanging feed dispensers and bell-type drinkers. The feeding program consisted of a starter diet (21% CP, 3200 kcal ME/kg DM) from 1 to 21 days of age and a finisher diet (19% CP, 3200 kcal ME/kg DM) from day 22 to 42. Each chicken was identified on the left leg with a numbered zip tie band of a different color for each treatment. The experiment was conducted between 21 and 42 days of age [14].

The target stocking densities at the end of the experiment were 30 kg/m^2^ (high stocking density), as recommended for commercial lines in hot weather conditions [16] and 15 kg/m^2^ for birds at low stocking densities [13]. An additional space of 0.25 m^2^/pen was provided to compensate for the area occupied by the feeder and drinker. Therefore, with the same number of birds per pen (10), the low stocking density pens had 5 birds/m^2^, with a treatment enclosure size of 2.25 m^2^ (1.5 m × 1.5 m), and the high stocking density pens had 10 birds/m^2^, with a treatment enclosure size of 1.25 m^2^ (1.5 m × 0.83 m). In each of the 3 buildings, there were four pens for each of the two stocking densities (Figure 1). For the outdoor access treatments, each pen had an outdoor area of dimensions 1.5 m × 6.7 m, whereby the birds could walk through a pop-hole of dimensions 0.50 m high × 1.50 m wide, via a ramp of 0.86 m × 1 m. Fences were made of galvanized chicken wire mesh to a height of 1.2 m. The 12 outdoor areas were covered with natural vegetation (mostly *Pennisetum ciliare* and *Leucaena leucocephala*) to a height of approximately 0.40 m [13,14].

### 2.3. Data Collection

#### 2.3.1. Behavior Recording

The birds in each season were observed by the same researcher (RS-C), following a purpose-designed ethogram of eight mutually exclusive behaviors (Table 1). The experimental pens were arranged so that the observer had visual access from the center of the building, without having to approach or move between pens. The numbers of birds engaged in each behavior were live-scored during the scan samples, taken at 10 min intervals for three 1 h periods per day (7:00–8:00; 12:00–13:00, and 16:00–17:00 h) and replicated once over the weeks 4 and 6 of the experimental period. For outdoor access treatments, the behavior was recorded regardless of whether the birds were inside or outside. The intra-observer reliability, determined from a previous observation in similar conditions [13], expressed as a Pearson correlation coefficient, was 0.99 [17]. The indoor and outdoor temperatures were measured with a portable weather station (AcuRite^®^ Weather Environment System, model 01057RM, Chaney Instrument Co., Lake Geneva, WI, USA) throughout the experimental period [13]. Indoor temperatures were measured at a height of 0.30 m, both at one of the two corners distant from the pop-holes and at the center of the pens of the outdoor access treatments.

#### 2.3.2. Stress Indicators

At 28 and 42 days old, for each season, 3 mL of blood was collected from the brachial vein of 18 birds per treatment (3 birds/pen). Samples were collected within one minute post-capture to avoid the handling effects on physiological parameters, after which birds were returned to their pens. Each sample was placed in a tube with anticoagulant. The whole blood was processed according to Fosoul et al. [18]. Smears were examined in oil immersion under a light microscope. The count of the various cell types was made on a total of 100 leukocytes. The absolute and relative blood cell counts were obtained. The absolute heterophil and lymphocyte counts were used to calculate heterophil to lymphocyte (H/L) ratios. Chickens were individually weighed before slaughter. The spleen and bursa of Fabricius of these same 18 animals/treatment/season were removed post-mortem and weighed. The relative weight (RW) of the organs to body weight was determined as a ratio [8,13].

#### 2.3.3. Leg Health

##### Footpad Dermatitis (FPD)

FPD was inspected at slaughter by one of us (R. S-C) in the same birds as previously described, using a five-point scale developed by the European consortium Welfare Quality^®^ [19]: 0: no lesions, 1: mild, 2: moderate, 3: severe, and 4: very severe lesions. The number of birds in each score category was subsequently combined into three classifications: a: no evidence of FPD (score 0), b: minimal evidence of FPD (score 1 and 2), and c: evidence of FPD (scores 3 and 4).

##### Tibia Bone Properties

The bone properties were inspected following slaughter. The soft tissues of the left tibias were removed, and the bone weight and length were measured, with the latter in a straight line, including both epiphyses, using a vernier calliper. The diaphysis diameter was measured in the narrowest part. The Seedor index was calculated by dividing the weight of the tibia by its length [20]. Finally, tibiae pieces were air-dried and washed using a muffle furnace at 600 °C for 16 h to determine the dry matter (DM) and ash content.

### 2.4. Statistical Analysis

The studies in summer and winter used different chickens in each season. The pen was the experimental unit for all analyses. A 2^4^ factorial design was used to examine the effects of season (winter or summer), outdoor access (with or without), stocking density (low: 5 animals/m^2^ or high: 10 animals/m^2^), and age (28 or 42 days) on the behavior and H/L ratio. As well as these main effects, entered as fixed effects, the statistical model included the following outdoor access interactions: season × outdoor access, season × stocking density, season × age, outdoor access × stocking density, outdoor access × age, stocking density × age, season × outdoor access × stocking density, season × outdoor access × age, season × stocking density × age, outdoor access × stocking density × age and season × outdoor access × stocking density × age. The relative weight of lymphoid organs and bone quality was analyzed using its 2^3^ factorial design, excluding “age” from the model. Behavior was analyzed as the percentage of time performing a given activity in each pen, with six observations at three times a day for each recording day (10 birds per pen × 6 intervals × 3 times a day × 2 days a week). The data were pooled, providing one value per behavior per pen. The assumption of ANOVA that residuals are normally distributed was tested using the Anderson–Darling test [13]. The dustbathing values were mathematically manipulated by taking square roots of the values to achieve normally distributed residuals. General linear models were applied to analyze data, using Minitab 17 (Minitab Inc., State College, PA, USA, 2014). A Tukey post-hoc test was used to discriminate significant differences between pairs of means when significant differences were detected overall [14]. Data for the behavior, H/L ratio, and relative weight of lymphoid organs are expressed as means, standard error of the difference between two means (SED), and *p*-values. For FPD, a Fisher’s exact test was used to measure differences between treatments and seasons. For this, contingency tables were constructed to contrast every treatment, season, and age. A *p*-value of ≤0.05 was considered significant in all analyses.

## 3. Results

### 3.1. Climate and Behavior

The average temperature and relative humidity recorded during the behavioral observations were normal for the tropical environment in which the study was conducted (Table 2, for recorded temperature and RH outside of these periods see [14]). Outdoor temperatures were only, on average, 1–3 °C hotter in summer than winter; however, the humidity was much greater in summer for the midday and afternoon recordings. Higher temperatures were commonly observed outdoors, compared with indoors. Indoor temperatures and humidity showed less variation than outdoor temperatures, with lower maximum and higher minimum values.

The main and interaction effects of season, outdoor access, stocking density, and age on the behavior of broilers are shown in Table 3, Table 4 and Table 5. Except for feeding, resting, and dustbathing, most behaviors were observed more frequently in summer, while resting was observed more frequently in winter. Chickens were more frequently observed walking or running (locomotion), preening, and foraging in low-density treatments, but birds in the high-density treatments drank less and spent more time resting. Providing outdoor access by itself did not affect any behavior, although there was a trend (*p* = 0.07) for resting to decrease when it was provided. Younger birds spent considerably more time in locomotion and resting and less standing, compared with older birds.

The interaction between bird age and outdoor access demonstrated that there were no differences between inside and outside for resting behavior in week 4, but resting behavior was higher for indoor birds than outdoor birds in week 6 (week 4 indoor 69.3%, week 4 outdoor 69.8%, week 6 indoor 63.3%, week 6 outdoor 59.1%, *p* = 0.02). This difference was only significant in summer, not winter. In summer, older birds with outdoor access rested less (46.4% of time) than those without outdoor access (54.4% of time; other treatments mean 70.4%, SED 2.03, *p* = 0.04). Similarly, in summer, low-density birds with outdoor access rested less (51.9%) than low-density birds without outdoor access (58.6%), whereas there was no difference between those with and without outdoor access at high densities (62.8 and 63.3%, respectively). In winter, there was no interaction between outdoor access and density (69.0–74.3%).

Season × outdoor access, season × density, and season × age interactions were significant for drinking, locomotion, resting, standing, preening, and foraging (Table 4), as described below.

Season × outdoor access interactions

In indoor birds, drinking increased in summer, but not in outdoor birds (Table 4). Birds in the outdoor treatments in summer were observed standing more frequently than those in winter, whereas there was no effect in indoor treatments.

Season × stocking density interactions

Drinking was more frequent at low densities in summer, compared to those at high densities in winter (Table 4). At low densities, locomotion was more frequently observed in summer, but there was no difference between the seasons at high densities. At high densities, locomotion was more frequently observed in summer, but there was no difference between the seasons at low densities. In summer, birds rested for longer in high densities, but there was no difference in winter. In winter, preening was more frequent in low densities, but in summer there was no difference. Foraging was greatest in birds at low densities in summer and lowest at high densities in winter.

Season × age interactions

Drinking increased in older birds in summer compared to younger birds in winter. Locomotion and standing increased in younger birds in summer, but not in older birds. Foraging increased in summer in older birds, but not in younger birds. Older birds rested more and preened and foraged less in winter.

Significant interactions between season, stocking density, and age are presented in Table 5. Birds raised in summer at low stocking densities drank more at 28 days old, compared with those in winter in high-density treatments at 28 and 42 days old. Locomotion was more frequently observed in younger chickens raised in summer, but only in low-density treatments, and older birds in the high-density treatments, in both winter and summer, showed less locomotion at 42 days old. Resting was higher in both younger and older birds raised in winter in high-density treatments, in contrast to older birds raised in summer in low-density treatments. Standing was only more frequently observed in older than younger birds when they were raised in winter at low stocking densities; apart from this, there was no difference between older and younger birds. Older birds raised in summer at high densities spent more time preening compared to older birds raised in winter in both low- and high-density treatments. Foraging was only more frequently observed in older than younger chickens when they were raised in summer at high densities; apart from this, there was no difference between older and younger birds.

### 3.2. Stress Indicators

#### 3.2.1. Heterophils and Lymphocytes

There were significant effects of season, stocking density, and age on heterophil and lymphocyte numbers, and H/L ratio (Table 6). Both heterophils and lymphocytes were greater in summer, but the H/L ratio was greater in winter. Heterophils were greater at low densities, and lymphocytes greater in older birds.

There were interactions between outdoor access and season (Table 7). Heterophil numbers were greater in the outdoor treatments in summer but not in winter, and were greater at low stocking densities in summer than at high densities in winter. Older birds increased heterophils in summer, but not younger birds, and lymphocytes also increased in summer.

The interaction between outdoor access, stocking density, and age (Table 8) showed that the heterophil number and H/L ratio were greater at low densities in younger birds with outdoor access. Lymphocytes were greater in older chickens raised with outdoor access at low densities, compared to younger birds raised with outdoor access at low densities and those raised in the indoor treatment at high densities.

#### 3.2.2. Lymphoid Organs and Tibial Properties

The effects of season, outdoor access, and stocking density on the relative weight (RW) of lymphoid organs are shown in Table 9 and Table 10. Both spleen and bursa of Fabricius were heavier in winter than summer (Table 9). In winter, spleens were heavier in indoor birds and those at low densities, but not in summer. In summer, the bursa of Fabricius was heavier in outdoor birds and those at low stocking densities, but not in winter (Table 10).

Tibiae were longer and narrower in summer, and longer and wider at low stocking densities (Table 11). The Seedor index increased at low stocking densities. Tibiae were also longer and tended to be narrower (*p* = 0.10) in the indoor birds compared with those with outdoor access. They tended to be heavier in summer at low stocking densities, but Tukey’s test did not indicate this as statistically significant. Significant interactions showed that tibiae were longest in indoor birds and those at low stocking densities in summer and shortest in outdoor birds and those at high stocking densities in winter (Table 12). In the outdoor system and at low stocking densities, tibiae were wider in winter than in summer, but in the indoor system there was no difference. The Seedor index for tibiae was greatest in birds at low stocking densities in winter and smallest in birds at high stocking densities in summer.

## 4. Discussion

This study was conducted with typical conditions for the tropics. Temperatures were high in both summer and winter, above the upper critical temperature, which has been estimated between 23.9 and 25.5 °C [21], but in summer the heat stress was exacerbated by high humidity. In our facility, the outdoor temperatures tended to be higher than those recorded indoors.

Outdoor access mainly affected resting behavior and tibiae conformation. A reduction in resting behavior was most evident in summer, both in older birds and in birds at low densities. The potential for heat stress in summer is sufficient to cause increased mortality in this climate [13,14]. The reduced resting time in outdoor birds in summer, compared with indoor birds, suggests that the high outdoor temperatures stimulated activity. This was most evident for birds at low densities and older birds. If outdoor access was sufficient to cause heat stress, it is possible that there was sufficient space for some subordinate birds to be forced by dominant birds to stay outside at low densities, where the high temperatures stimulated activity and reduced growth and feed conversion efficiency [14]. High-density indoors would probably offer less opportunity for agonistic bird interactions.

Low densities had welfare benefits of encouraging more activity in the birds, locomotion, preening, and foraging, and high densities encouraged polydipsia and reduced activity. The decreased drinking in birds at high densities in summer may have been connected to the increased resting, standing, and locomotion of these birds at this time. The close proximity of other birds is likely to have stimulated locomotion and standing. This in turn would have necessitated more time spent resting. Conversely, birds at low densities spent more time foraging, conceivably because they were less stressed, and in winter they spent more time preening, probably because of reduced disturbances by other birds.

As expected, drinking behavior was particularly high in week 6 of the experimental period (42 d old birds) in summer, when high temperatures were recorded. This is in accordance with the findings of Bruno et al. [22], who reported that high environmental temperatures increase this behavior. The bell-type drinkers used in our study promoted high water intake, which is important for broilers in terms of biochemical and physiological function for homeostasis and growth.

The resting behavior observed more frequently in both outdoor access treatments and densities during winter is in accordance with the findings of Sanchez-Casanova et al. [13], demonstrating a dependency of this response on season. Since resting is a comfort behavior, chickens spend up to 76% of their time in it [23], but it can be a response to restricted space or leg disease [24,25]. However, despite the reduced percentage of foot pad dermatitis observed in winter, the fact that chickens at 42 days old were heavier in low-density treatments, with inferior tibia bone development at this age, could have led to the reduced physical activity, decreasing locomotion and increasing resting behavior.

Standing behavior followed a pattern that could be explained by the opposite effect described for resting, with similar consequences derived from bone quality. This behavior was more frequently observed in summer and the effects of stocking density were also dependent on season since the high- and low-density treatments showed an opposite trend. In summer, more birds were found standing in the high-density treatments. In terms of age, there was also an opposite trend in winter, since more birds were observed standing at 42 days old, in contrast to those at 28 days old. Standing has been described as a sign of good leg health, but is also considered as an avoidance response when discomfort is present [26], and it can also be a response to heat stress.

There was a higher percentage of FPD lesions, poorer tibia bone properties, and the hardest environmental conditions observed in summer, which could be increased by high stocking densities [27]. The pain produced by FPD ulcerative lesions in conjunction with high temperatures and humidity observed in week 4 of the experimental period in summer (Table 2) could have increased the frequency of this avoidance behavior (standing), which was exacerbated by acute and chronic pain derived from an impaired leg bone development [23,26].

More birds were observed preening in summer, at both low and high stocking densities, at 28 and 42 days old, but with no difference between outdoor and indoor treatments. Although birds usually prefer to preen themselves indoors [28], increased preening has been related with mild frustration [29] which could be provoked by thermal discomfort in summer. However, the fact that this behavior increased at the end of the experimental period agrees with the findings of Fortomaris et al. [30], who suggested that most husbandry systems allow full expression of this behavior.

Foraging is a highly motivated behavior [31] and is a good indicator of a comfortable state in chickens. In this study, low-density treatments promoted foraging in birds raised in summer which can be attributable to more space allowance and, therefore, less effects of hard environmental conditions. In addition, litter material could have served as a source of enrichment, leading to a less restricted natural behavior.

Heterophils increased in summer at 42 days old, and the higher H/L ratio observed in winter in the LO treatment, at 28 days old, as a result of heterophilia, was contrary to the findings of Osti et al. [32], whereby the H/L ratio was greater during summer, in accordance with Sanchez-Casanova et al. [13], who found a higher H/L ratio in low-density treatments in winter, and Rajkumar et al. [33], whose findings demonstrated a lower H/L ratio in summer. An increase in the H/L ratio is commonly observed during mild to moderate stress [34,35] and is also associated with elevated serum corticosterone concentrations [36]. Multiple stressors can increase this ratio such as fearfulness and heat stress. Although fearfulness was not measured in this study, it is well known that handling procedures and external factors, such as noise or predators, could trigger fear, primarily around young ages due to little experience being outside [37]. Heat stress is often observed when metabolic heat production is greater than the amount of heat that can be lost to the environment, so body heat and body temperature increase [38]. The duration of heat exposure plays an important role in the H/L ratio [39]. Even though outdoor access could potentially improve chickens’ welfare by providing extra space, it could also expose chickens to high temperatures [40]. Tropical environmental temperatures over 30 °C, as observed during the week 4 of the experimental period in winter and summer (Table 2), could have led to thermal discomfort. There was a rise in the H/L ratio after heat exposure over 6 h [39]; the 11 h of sunlight exposure in our outdoor areas could have caused this response, especially in commercial broilers, which are less tolerant to heat stress [13,40].

Decreased spleen and bursa RW observed indoors in summer, as well as decreased bursa RW in high-density treatments, are in accordance with the H/L results discussed above and with previous studies [7,8], which confirm that decreased RW of lymphoid organs is associated with chronically elevated corticosterone concentrations. This elicits the involution of lymphoid organs, such as the spleen and bursa of Fabricius, by the depletion of lymphocytes from germinal cells with dysregulated immune responses [7,41,42]. However, other researchers have found decreased weight of the bursa of Fabricius as a result of heat stress [41]. On the other hand, an increased bursa of Fabricius RW, as observed in winter and at low densities, could indicate the maturation of a greater number of T lymphocytes and B lymphocytes, since the bursa provides a suitable microenvironment for the proliferation and differentiation of these cells [42,43]. The larger the spleen, the stronger the immune system. However, the avian spleen size cannot always be used as an indicator of optimal immune status, as it has large seasonal intra-individual variations [44].

The decreased resting among birds with outdoor access may have been responsible for their different tibia shape. The tibiae were wider and tended to be shorter, suggesting greater activity. Physical activity increases the diameter of the tibiotarsus diaphysis [45]. This may be beneficial for tibiae strength, with breakages reflecting a potential problem when birds are caught. The better tibia bone properties observed in the LI treatment confirm that low stocking densities promoted behaviors such as walking and running (locomotion), which increased both the thickness and density of the cortical bone, as well as the diameter of the diaphysis [45], as indicated by the higher Seedor index at low densities in winter. However, as physical activity tends to decrease with age this study and [28], in combination with the consequences of the fast growth rate in broilers [23], tibia bone properties could also be reduced over time.

## 5. Conclusions

Outdoor access had variable effects on birds’ behavior and leg health, depending on season, bird age, and stocking density indoors. In summer, outdoor access reduced resting behavior, especially in older birds and those at low densities inside. This may have been associated with increased heterophils, which were observed to increase in outdoor treatments in summer, but not winter, and particularly in birds at low stocking densities inside. The other major effect of outdoor access was to increase the width of the birds’ tibiae. This is likely to have been a response to increased activity outdoors. The results of this study emphasize that there can be risks caused by reducing resting when providing outdoor access during the summer months in the tropics.

## Figures and Tables

**Figure 1 animals-12-01917-f001:**
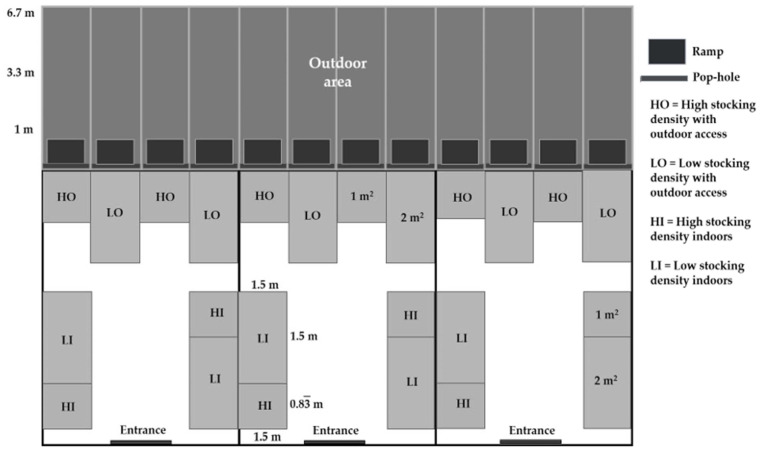
Experimental poultry facilities (from Sanchez-Casanova et al. [14]).

**Table 1 animals-12-01917-t001:** Ethogram of mutually exclusive behaviors.

Category	Behavior	Description
Individual	Feeding	Eating from food hopper, whilst standing, sitting, or resting.
	Drinking	Drinking from the water trough, whilst standing, sitting, or resting.
	Locomotion	Moving by walking or running.
	Resting	Main part of the body touching the ground, either chest or side.
	Standing	The abdomen not touching the litter or ground and the bird is motionless with no apparent movement of legs.
	Preening	Moving beak along the plumage.
Interaction with the environment	Dustbathing	While lying with fluffed feathers, the bird simultaneously and rapidly lifts its wings up and down multiple times, while scooping the loose substrate material up into its feathers.
	Foraging	Scratching at the ground, both inside the pen and outdoors, with intermittent bouts of ground pecking at items (visible or not), usually followed by one or two steps backwards after a bout of ground scratching.

**Table 2 animals-12-01917-t002:** Dry bulb temperature (°C) and relative humidity (RH, %) with standard deviations recorded during the experimental period (abridged from Sanchez-Casanova et al. [14]).

Hour	Outdoors				Indoors			
	Min (°C)	Max (°C)	Average (°C)	RH (%)	Min (°C)	Max (°C)	Average (°C)	RH (%)
Winter								
Week 4								
7:00–8:00	21	24	22.8 ± 1.03	88.1 ± 2.23	23	24	23.8 ± 0.45	84.6 ± 1.16
12:00–13:00	32	35	33.7 ± 0.78	46.7 ± 5.53	31	32	30.9 ± 0.67	52.5 ± 7.45
16:00–17:00	31	33	32.1 ± 0.67	47.6 ± 3.23	30	31	30.7 ± 0.49	50.3 ± 1.23
Week 6								
7:00–8:00	19	23	20.9 ± 1.38	89.1 ± 1.68	21	24	22.8 ± 1.14	86.6 ± 2.11
12:00–13:00	32	36	34.4 ± 1.56	49.8 ± 2.99	31	33	31.7 ± 0.78	57.2 ± 3.01
16:00–17:00	27	30	28.7 ± 1.23	68.5 ± 3.18	27	30	28.7 ± 0.98	68.5 ± 3.37
Summer								
Week 4								
7:00–8:00	22	25	23.2 ± 1.17	89.5 ± 0.55	25	28	25.3 ± 0.52	86.5 ± 0.55
12:00–13:00	34	38	34.7 ± 0.52	62.8 ± 2.93	32	36	32.8 ± 0.75	65.7 ± 3.33
16:00–17:00	35	36	35.3 ± 0.49	65.5 ± 2.88	34	35	34.0 ± 0.01	61.8 ± 2.04
Week 6								
7:00–8:00	24	28	25.7 ± 1.63	86.8 ± 3.37	27	29	27.7 ± 0.82	82.2 ± 1.60
12:00–13:00	34	36	36.8 ± 0.75	54.0 ± 1.26	33	35	34.5 ± 0.55	55.7 ± 1.21
16:00–17:00	32	34	26.7 ± 3.06	82.8 ± 4.37	30	33	29.7 ± 2.31	74.2 ± 3.59

Weeks 4 and 6 of each season corresponded to the age of birds at 28 and 42 days old, respectively.

**Table 3 animals-12-01917-t003:** The effects of season, outdoor access, stocking density, and age on the behavior of Cobb500 broilers ^1^.

	Feeding	Drinking	Locomotion	Resting	Standing	Preening	Dustbathing	Foraging
Season								
Winter	13.2	5.63	0.60	71.6	3.87	3.52	0.61	0.89
Summer	13.1	7.16	1.11	59.2	4.93	5.50	0.53	1.50
*p*-value	0.46	0.003	<0.001	<0.001	0.001	<0.001	0.89	0.001
SED	0.72	0.49	0.11	1.01	0.31	0.35	0.24	0.18
Outdoor access								
With	13.0	6.39	0.89	64.5	4.45	4.39	0.46	1.23
Without	13.6	6.40	0.80	66.3	4.35	4.63	0.68	1.16
*p*-value	0.20	0.97	0.43	0.07	0.75	0.50	0.23	0.69
SED	0.72	0.49	0.11	1.01	0.31	0.35	0.24	0.18
Stocking density								
Low	13.0	7.06	1.31	62.6	3.32	5.03	0.71	1.50
High	13.3	5.74	0.38	68.2	5.48	3.99	0.44	0.89
*p*-value	0.63	0.009	<0.001	<0.001	<0.001	0.004	0.31	0.001
SED	0.72	0.49	0.11	1.01	0.31	0.35	0.24	0.18
Age								
28 d	13.0	6.22	1.19	69.6	3.81	4.54	0.44	1.12
42 d	13.4	6.57	0.50	61.2	4.98	4.48	0.70	1.27
*p*-value	0.49	0.47	<0.001	<0.001	<0.001	0.86	0.99	0.39
SED	0.72	0.49	0.11	1.01	0.31	0.35	0.24	0.18
Interactions, *p*-values								
S × O	0.40	0.03	0.001	<0.001	0.02	<0.001	0.13	0.03
S × D	0.12	0.001	0.001	0.03	0.04	<0.001	0.87	<0.001
S × A	0.17	0.04	0.04	<0.001	0.001	<0.001	0.72	0.003
O × D	0.86	0.20	0.68	0.62	0.87	0.28	0.31	0.33
O × A	0.68	0.70	0.68	0.02	0.41	0.99	0.81	0.64
D × A	0.85	0.16	0.02	0.32	0.33	<0.001	0.36	0.002
S × O × D	0.38	0.84	0.69	0.01	0.98	0.91	0.83	0.29
S × O × A	0.32	0.68	0.99	0.04	0.04	0.02	0.63	0.58
S × D × A	0.65	0.01	0.04	<0.001	0.03	0.001	0.06	0.01
O × D × A	0.08	0.55	0.31	0.97	0.75	0.10	0.96	0.82
S × O × D × A	0.39	0.99	0.55	0.16	0.08	0.06	0.46	0.98

^1^ Percentage of time performing each behavior, based on a total of 360 min of observation. S = season, O = outdoor access, D = stocking density, A = age, SED = standard error of the difference. Data for dustbathing were square-root-transformed for ANOVA, but means provided were those calculated from original data.

**Table 4 animals-12-01917-t004:** The effects of interactions between season and outdoor access, density, and age on the behavior of Cobb500 broilers ^1^.

	Outdoor Access		Stocking Density		Age	
	With	Without	Low	High	28 d	42 d
Feeding						
Summer	12.4	13.9	12.4	13.9	12.4	13.9
Winter	13.0	13.3	13.6	12.8	13.4	12.9
SED	1.01		1.01		1.01	
*p*-value	0.40		0.12		0.17	
Drinking						
Summer	6.94 ^ab^	7.38 ^a^	7.94 ^a^	6.34 ^ab^	6.97 ^ab^	7.36 ^a^
Winter	5.83 ^ab^	5.43 ^b^	6.18 ^ab^	5.08 ^b^	5.47 ^b^	5.79 ^ab^
SED	0.70		0.70		0.70	
*p*-value	0.03		0.001		0.04	
Locomotion						
Summer	1.18 ^a^	1.04 ^a^	1.78 ^a^	0.44 ^bc^	1.57 ^a^	0.65 ^bc^
Winter	0.60 ^b^	0.56 ^b^	0.83 ^b^	0.32 ^c^	0.81 ^b^	0.35 ^c^
SED	0.16		0.16		0.16	
*p*-value	0.001		0.001		0.04	
Resting						
Summer	57.4 ^b^	61.0 ^b^	55.3 ^c^	63.1 ^b^	68.0 ^b^	50.4 ^c^
Winter	71.6 ^a^	71.6 ^a^	70.0 ^a^	73.2 ^a^	71.2 ^ab^	72.0 ^a^
SED	1.43		1.43		1.43	
*p*-value	<0.001		0.03		<0.001	
Standing						
Summer	5.09 ^a^	4.77 ^ab^	3.52 ^bc^	6.34 ^a^	4.88 ^a^	4.98 ^a^
Winter	3.81 ^b^	3.93 ^ab^	3.12 ^c^	4.62 ^b^	2.74 ^b^	5.00 ^a^
SED	0.44		0.44		0.44	
*p*-value	0.02		0.04		0.001	
Preening						
Summer	5.35 ^a^	5.65 ^a^	5.88 ^a^	5.11 ^ab^	4.35 ^b^	6.64 ^a^
Winter	3.44 ^b^	3.61 ^b^	4.19 ^b^	2.86 ^c^	4.73 ^b^	2.31 ^c^
SED	0.50		0.50		0.50	
*p*-value	<0.001		<0.001		<0.001	
Dustbathing						
Summer	0.25	0.81	0.62	0.44	0.44	0.62
Winter	0.66	0.56	0.78	0.43	0.45	0.77
SED	0.34		0.34		0.34	
*p*-value	0.13		0.87		0.72	
Foraging						
Summer	1.48 ^a^	1.53 ^a^	1.78 ^a^	1.23 ^ab^	1.27 ^ab^	1.74 ^a^
Winter	0.98 ^ab^	0.79 ^b^	1.21 ^ab^	0.56 ^b^	0.96 ^b^	0.81 ^b^
SED	0.26		0.26		0.26	
*p*-value	0.03		<0.001		0.003	

^1^ Percentage of time performing each behavior, based on a total of 360 min of observation. SED = standard error of the difference. Data for dustbathing were square-root-transformed for ANOVA but means provided were calculated from the original data. Means of the same behavior and same factor with different superscript letters indicate statistically significant differences (Tukey’s HSD test; *p* ≤ 0.05).

**Table 5 animals-12-01917-t005:** Effects of the interaction between season, stocking density, and age on the behavior of Cobb 500 broilers ^1^.

			Feeding	Drinking	Locomotion	Resting	Standing	Preening	Dustbathing	Foraging
Season	Density	Age								
		28 d	13.7	5.99 ^ab^	1.13 ^b^	69.1 ^ab^	2.48 ^c^	5.55 ^ab^	0.69	1.34 ^ab^
	Low									
		42 d	13.4	6.36 ^ab^	0.53 ^bc^	70.8 ^ab^	5.14 ^a^	2.82 ^c^	0.88	1.09 ^ab^
Winter										
		28 d	13.1	4.95 ^b^	0.49 ^bc^	73.2 ^a^	3.01 ^bc^	3.91 ^bc^	0.21	0.58 ^b^
	High									
		42 d	12.4	5.21 ^b^	0.16 ^c^	73.2 ^a^	4.86 ^ab^	1.80 ^c^	0.65	0.54 ^b^
		28 d	11.9	8.47 ^a^	2.54 ^a^	65.5 ^b^	5.51 ^a^	6.11 ^ab^	0.23	2.08 ^a^
	Low									
		42 d	12.9	7.40 ^ab^	1.02 ^b^	45.0 ^d^	4.67 ^ab^	5.65 ^ab^	1.02	1.48 ^ab^
Summer										
		28 d	12.9	5.46 ^ab^	0.60 ^bc^	70.4 ^ab^	4.26 ^abc^	2.59 ^c^	0.65	0.46 ^b^
	High									
		42 d	12.9	7.31 ^ab^	0.28 ^c^	55.8 ^c^	5.28 ^a^	7.64 ^a^	0.23	1.99 ^a^
SED			1.43	0.98	0.23	2.03	0.63	0.71	0.48	0.36
*p*-value			0.65	0.01	0.04	<0.001	0.03	0.001	0.06	0.01

^1^ Percentage of time performing each behavior, based on a total of 360 min of observation. SED = standard error of the difference. Data for dustbathing were square-root-transformed for ANOVA, but values provided were calculated from original data. Means in the column of each behavior with different superscript letters indicate statistically significant differences (Tukey’s HSD test; *p* ≤ 0.05).

**Table 6 animals-12-01917-t006:** Effects of season, outdoor access, stocking density, and age on the heterophils, lymphocytes, and H/L ratio of Cobb500 broilers ^1^.

	Heterophils (%) *	Lymphocytes (%) *	H/L Ratio
Season			
Winter	37.8	40.9	1.06
Summer	42.9	46.3	0.95
*p*-value	0.001	<0.001	0.04
SED	0.97	0.88	0.05
Outdoor access			
With	41.4	43.9	0.99
Without	41.3	43.3	1.03
*p*-value	0.87	0.44	0.49
SED	0.97	0.88	0.05
Stocking density			
Low	42.5	43.5	1.04
High	40.2	43.8	0.98
*p*-value	0.02	0.75	0.22
SED	0.97	0.88	0.05
Age			
28 d	40.5	41.6	1.03
42 d	42.1	45.7	0.99
*p*-value	0.10	<0.001	0.41
SED	0.97	0.88	0.05
Interactions, *p*-value			
S × O	0.56	0.56	0.38
S × D	0.89	0.38	0.54
S × A	0.07	0.43	0.40
O × D	0.08	0.27	0.20
O × A	0.95	0.80	0.63
D × A	0.20	0.72	0.30
S × O × D	0.20	0.35	0.31
S × O × A	0.83	0.31	0.43
S × D × A	0.28	0.85	0.69
O × D × A	0.02	0.07	0.02
S × O × D × A	0.15	0.14	0.07

^1^ Based on a total of 72 samples at 28 days old and 68 at 42 days old. * Of the total white blood cell count. S = season, O = outdoor access, D = stocking density, A = age, SED = standard error of the difference. Means of the same variable and same factor with different superscript letters indicate statistically significant differences (Tukey’s HSD test; *p* ≤ 0.05).

**Table 7 animals-12-01917-t007:** Effects of season × system, season × density, and season × age interactions on the heterophils, lymphocytes, and H/L ratio of Cobb500 broilers ^1^.

	Outdoor Access		Stocking Density		Age	
	With	Without	Low	High	28 d	42 d
Heterophils (%) *						
Summer	43.3 ^a^	42.5 ^ab^	44.1 ^a^	41.7 ^ab^	41.2 ^ab^	44.6 ^a^
Winter	39.6 ^b^	40.0 ^ab^	40.8 ^ab^	38.7 ^b^	39.9 ^b^	39.7 ^b^
SED	1.37		1.37		1.38	
*p*-value	0.04		<0.001		0.002	
Lymphocytes (%) *						
Summer	46.4 ^a^	46.3 ^a^	45.8 ^a^	46.9 ^a^	44.7 ^b^	48.0 ^a^
Winter	41.5 ^b^	40.3 ^b^	41.2 ^b^	40.7 ^b^	38.6 ^c^	43.3 ^b^
SED	1.25		1.23			
*p*-value	<0.001		<0.001		<0.001	
H/L ratio						
Summer	0.96	0.95	1.00	0.91	0.96	0.96
Winter	1.02	1.10	1.07	1.04	1.10	1.02
SED	0.07		0.07		0.07	
*p*-value	0.16		0.09		0.16	

^1^ Based on a total of 72 samples at 28 days old and 68 at 42 days old. * Of the total white blood cell count. SED = standard error of the difference. Means of the same variable and same factor with different superscript letters indicate statistically significant differences (Tukey’s HSD test; *p* ≤ 0.05).

**Table 8 animals-12-01917-t008:** Effects of the interaction between season, stocking density, and age on the heterophils, lymphocytes, and H/L ratio of Cobb500 broilers ^1^.

Outdoor Access	Density	Age	Heterophils (%) *	Lymphocytes (%) *	H/L Ratio
		28 d	44.4 ^a^	40.5 ^bc^	1.17 ^a^
	Low				
		42 d	42.4 ^ab^	46.2 ^a^	0.94 ^ab^
With					
		28 d	36.9 ^b^	43.6 ^abc^	0.88 ^b^
	High				
		42 d	41.9 ^ab^	45.6 ^ab^	0.98 ^ab^
		28 d	40.2 ^ab^	42.1 ^abc^	1.00 ^ab^
	Low				
		42 d	42.9 ^ab^	45.2 ^abc^	1.05 ^ab^
Without					
		28 d	40.6 ^ab^	40.2 ^c^	1.07 ^ab^
	High				
		42 d	41.3 ^ab^	45.7 ^ab^	0.99 ^ab^
SED			1.88	1.81	0.09
*p*-value			0.002	0.02	0.04

^1^ Based on a total of 72 samples at 28 days old and 68 at 42 days old. * Of the total white blood cell count. SED = standard error of the difference. Means in the column of each variable and rows of each factor with different superscript letters indicate statistically significant differences (Tukey’s HSD test; *p* ≤ 0.05).

**Table 9 animals-12-01917-t009:** Effects of season, outdoor access, and stocking density on the relative weight of lymphoid organs of Cobb500 broilers.

	Spleen RW (%)	Bursa of Fabricius RW (%)
Season		
Winter	0.12	0.14
Summer	0.09	0.08
*p*-value	<0.001	<0.001
SED	0.006	0.005
Outdoor access		
With	0.10	0.11
Without	0.11	0.11
*p*-value	0.14	0.15
SED	0.006	0.005
Stocking density		
Low	0.10	0.11
High	0.11	0.11
*p*-value	0.30	0.17
SED	0.006	0.005
Interactions, *p*-value		
S × O	<0.001	0.01
S × D	0.01	0.001
O × D	0.31	0.27
S × O × D	0.06	0.65

S = season, O = outdoor access, D = stocking density, SED = standard error of the difference. Means in the same column and same factor with different superscript letters indicate statistically significant differences (Tukey’s HSD test; *p* ≤ 0.05).

**Table 10 animals-12-01917-t010:** Effects of season × system and season × density interactions on the relative weight of lymphoid organs of Cobb500 broilers.

	Outdoor Access		Stocking Density	
	With	Without	Low	High
Spleen RW (%)				
Summer	0.10 ^b^	0.10 ^b^	0.10 ^b^	0.11 ^ab^
Winter	0.11 ^b^	0.14 ^a^	0.12 ^a^	0.12 ^a^
SED	0.008		0.008	
*p*-value	<0.001		0.01	
Bursa of Fabricius RW (%)				
Summer	0.09 ^b^	0.07 ^c^	0.10 ^b^	0.07 ^c^
Winter	0.13 ^a^	0.14 ^a^	0.13 ^a^	0.14 ^a^
SED	0.007		0.007	
*p*-value	0.01		0.001	

SED = standard error of the difference. Means of same variable and same factor with different superscript letters indicate statistically significant differences (Tukey’s HSD test; *p* ≤ 0.05).

**Table 11 animals-12-01917-t011:** Effects of season, outdoor access, and stocking density on the tibia bone properties of Cobb500 broilers.

	Weight (g)	Length (cm)	Diameter (cm)	Seedor Index	DM (%)	Ash Content (%)
Season						
Winter	19.1	10.6	0.83	1.79	53.5	39.6
Summer	19.3	11.1	0.78	1.73	53.1	39.1
*p*-value	0.64	<0.001	0.003	0.07	0.60	0.48
SED	0.43	0.08	0.02	0.03	0.75	0.65
Outdoor access						
Outdoor access	19.1	10.7	0.82	1.78	53.4	39.1
Indoor only	19.3	11.0	0.79	1.75	53.2	39.5
*p*-value	0.66	0.002	0.10	0.40	0.70	0.58
SED	0.43	0.08	0.02	0.03	0.75	0.65
Stocking density						
Low	20.2	11.0	0.84	1.82	53.4	39.1
High	18.2	10.6	0.77	1.71	53.3	39.5
*p*-value	<0.001	<0.001	<0.001	0.004	0.97	0.54
SED	0.50	0.09	0.02	0.04	0.75	0.65
Interactions, *p*-value						
S × O	0.33	0.81	0.10	0.27	0.15	0.18
S × D	0.59	0.25	0.13	0.90	0.21	0.55
O × D	0.001	0.54	0.68	<0.001	0.004	0.64
S × O × D	0.06	0.07	0.29	0.12	0.81	0.28

S = season, O = outdoor access, D = stocking density, SED = standard error of the difference, DM = dry matter. Means in the same column and same factor with different superscript letters indicate statistically significant differences (Tukey’s HSD test; *p* ≤ 0.05).

**Table 12 animals-12-01917-t012:** Effects of season × system and season × density interactions on the tibia bone properties of Cobb500 broilers.

	Outdoor Access		Stocking Density	
	Outdoor	Indoor	Low	High
Weight (g)				
Summer	19.4	19.2	20.4 ^a^	18.2 ^b^
Winter	18.8	19.4	19.9 ^a^	18.2 ^b^
SED	0.60		0.71	
*p*-value	0.33		0.01	
Length (cm)				
Summer	10.9 ^ab^	11.2 ^a^	11.3 ^a^	10.8 ^b^
Winter	10.5 ^c^	10.7 ^bc^	10.8 ^b^	10.4 ^c^
SED	0.12		0.13	
*p*-value	<0.001		<0.001	
Diameter (cm)				
Summer	0.78 ^b^	0.78 ^b^	0.80 ^b^	0.76 ^b^
Winter	0.86 ^a^	0.80 ^ab^	0.88 ^a^	0.79 ^b^
SED	0.02		0.02	
*p*-value	0.008		<0.001	
Seedor index (g/cm)				
Summer	1.77	1.70	1.79 ^ab^	1.70 ^b^
Winter	1.79	1.80	1.85 ^a^	1.74 ^ab^
SED	0.04		0.05	
*p*-value	0.17		0.005	
Dry matter (%)				
Summer	52.4	53.8	52.7	53.6
Winter	54.0	53.1	54.0	53.1
SED	1.07		1.07	
*p*-value	0.49		0.79	
Ash content (%)				
Summer	38.5	39.7	38.7	39.5
Winter	39.8	39.3	39.6	39.6
SED	0.92		0.92	
*p*-value	0.18		0.55	

SED = standard error of the difference. Means in the same column and same factor with different superscript letters indicate statistically significant differences (Tukey’s HSD test; *p* ≤ 0.05).

## Data Availability

The data presented in this study are available on request from the corresponding author.

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
