# Peer review of "The Effects of Providing Outdoor Access to Broilers in the Tropics on Their Behaviour and Stress Responses"

_animals, 2022, doi:10.3390/ani12151917_

Round 1

Reviewer 1 Report

The manuscript submitted for review presents an interesting study on the effects of outdoor access on behavior and stress indicators in broilers.

I have some comments, that are as follows:

Material and methods, sections 2.3.3.1 and 2.3.3.2 - were the samples collected from the same birds as in 2.3.2?

Description of tables e.g. 7 states that there were 72 samples at day 28, and 68 samples at day 42 - why?

Descroption of tables: * Over the total white blood cell count - there is no * indicated in the tables, moreover the statement is not clear, do you mean percentage of the total WBC?

I do not understand indications of statistically significant differences in the tables (eg. Table 4, 5, 7, 8...), I cant't see which values differ

Reviewer 2 Report

The paper examines the effects of broilers outdoor access on behaviour and stress responses in the tropics. The topic is relevant since there is a growing demand for poultry meat production worldwide. The paper presents novel and useful findings. The introduction provides evidence-based background for the research. The methods have been properly described, results are well presented and data interpretation is appropriate. The findings are thoroughly discussed, and conclusions are justified by the results but the conclusion can be improved. I did not find any objective errors, except those one marked in the text. 

I recommend publication of the article after minor corrections.

All the best and stay safe

Reviewer 3 Report

Lines 245-246 “but birds in the high-density treatments drank more and spent more time resting.” Not true for drinking : Low 7.06, high 5.74.

Line 271 change “was increased in winter” as “was increased in Summer”

Lines 282-283 “Drinking was more frequent in the Low density in summer, compared to those in  the High density in winter (Table 4).” in the Table 4 the letters are the same (a vs ab), check

Lines 283-285 “In High density, locomotion and standing were more frequently observed in summer than winter, but in Low density there were no differences between seasons” rewrite in accordance with the letters shown in Table 4.

Lines 287- 288 “Foraging was greatest in birds in Low density in summer and least in High density in winter” rewrite in accordance with the letters shown in Table 4.

Lines 291-294 “Drinking was increased in older birds in summer compared to younger birds in win-  ter. Locomotion, standing, and foraging were increased in summer in younger birds, but not in older birds. Older birds rested more and preened less in winter but not younger birds.” rewrite in accordance with the letters shown in Table 4.

Lines 307-309 “Standing was more frequently observed in older and younger birds in summer, in both Low and High-density treatments, at 28 and 42 days old, in contrast to younger birds in winter, at Low density.” check the letters shown in the table 2.48c vs 4.26abc.

Lines 311-313 “Foraging was more frequently observed in younger chickens  raised in summer, at Low density, in contrast to those raised at high density, in both winter and summer.” true only for the summer.

Line 320 “and lymphocytes greater in older birds” Missing letters in the table.

Lines 321-322 “Heterophil numbers were greater in the Outdoor treatments in summer but not in winter” in the Table 7 the letters are the same (a vs ab), check.

Line 325 “but particularly in the younger birds” delete.

Lines 335-337 “Lymphocytes were greater in older chickens raised with Outdoor access at low density, compared to younger birds raised in the Indoor treatment at high density” They are also different from younger birds raised with outdoor access at low density.

Tables 9-11 missing superscript letters.

Line 461 after “mild to moderate stress [34]” add 35

35 -  Hosseini-Vashan, S.J.; Piray, A.H. Effect of dietary saffron (Crocus sativus) petal extract on growth performance, blood biochemical indices, antioxidant balance, and immune responses of broiler chickens reared under heat stress conditions. It. J. Anim. Sci. 2021, 20, 1338-1347

https://doi.org/10.1080/1828051X.2021.1921628

Reviewer 4 Report

This paper looked at the effects of providing outdoor access to broilers in the tropics on behavior and stress response. They used 240 Cobb500 broilers which were placed into 4 treatment groups: low stocking density with no outside access, low stocking density with outside access, high stocking density with no outside access, and high stocking density with outside access with six replicates of 10 birds in each. It was conducted in the winter and the summer. For the behavior analysis, they used scan sampling, taken at 10-minute intervals for three 1-hour periods per day, and replicated once over the weeks 4 and 6 of the experimental period. Looked at feeding, drinking, locomotion, resting, standing, preening, dustbathing, and foraging. On days 28 and 42 days old, blood samples were collected to assess heterophil to lymphocyte ratio. At 42 days old, they chickens were culled and examined for foot pad dermatitis and bone quality. Their spleens and bursas were collected and weighed, relative to carcass weight. There were no major effects of provision of outdoor access on behavior, except that resting was reduced by providing outdoor access to older birds and those at low stocking density. Resting was also greater in indoor- and high-density treatments during the winter. The bursa was heavier in the summer in outdoor birds, and the tibia bones were shorter in the outdoor birds. Heterophil numbers were greater in the outdoor treatments in the summer but not in the winter.  These results indicate that outdoor access can increase activity in some situations, and potentially increase bone strength, but it may also increase the risk of heat stress in the summer.

CONCERNS

1.     Are you able to confidently say that the bones of the broilers were stronger after only 21 days? Why only compare bones from broilers starting at 21 days?

2.     Increase intellectual discussion of the differences in organ weights, larger bursa and spleen could be indicative of good things. Since they are exposed to more antigens while outdoors, their spleen and bursas would be expected to be larger.

3.     The authors need to go into more detail about how they actually conducted the behavior observations. Did the observer count how many ducks were performing each behavior every 10 minutes?  Was this live scoring or video?   If live scoring then how did the observer affect the behaviors?

4.     Can you have intra-observer probability if there was only one observer? Or were there multiple?

5.     Were the broilers allowed outside access 24/7?   Were they put indoors at night?  If so, how did that affect the density of birds during the scotophase?
